# NETWORK PRUNING THAT MATTERS:
# A CASE STUDY ON RETRAINING VARIANTS

**Duong H. Le**
VinAI Research, Vietnam

**Binh-Son Hua**
VinAI Research and VinUniversity, Vietnam

## ABSTRACT

Network pruning is an effective method to reduce the computational expense of over-parameterized neural networks for deployment on low-resource systems. Recent state-of-the-art techniques for retraining pruned networks such as weight rewinding and learning rate rewinding have been shown to outperform the traditional fine-tuning technique in recovering the lost accuracy (Renda et al., 2020), but so far it is unclear what accounts for such performance. In this work, we conduct extensive experiments to verify and analyze the uncanny effectiveness of learning rate rewinding. We find that the reason behind the success of learning rate rewinding is the usage of a *large* learning rate. Similar phenomenon can be observed in other learning rate schedules that involve large learning rates, e.g., the 1-cycle learning rate schedule (Smith & Topin, 2019). By leveraging the *right* learning rate schedule in retraining, we demonstrate a counter-intuitive phenomenon in that randomly pruned networks could even achieve better performance than methodically pruned networks (fine-tuned with the conventional approach). Our results emphasize the cruciality of the learning rate schedule in pruned network retraining – a detail often overlooked by practioners during the implementation of network pruning.

## 1 INTRODUCTION

Training neural networks is an everyday task in the era of deep learning and artificial intelligence. Generally speaking, given data availability, large and cumbersome networks are often preferred as they have more capacity to exhibit good data generalization. In the literature, large networks are considered easier to train than small ones (Neyshabur et al., 2018; Arora et al., 2018; Novak et al., 2018; Brutzkus & Globerson, 2019). Thus, many breakthroughs in deep learning are strongly correlated to increasingly complex and over-parameterized networks.

However, the use of large networks exacerbate the gap between research and practice since real-world applications usually require running neural networks in low-resource environments for numerous purposes: reducing memory, latency, energy consumption, etc. To adopt those networks to resource-constrained devices, network pruning (LeCun et al., 1990; Han et al., 2015; Li et al., 2016) is often exploited to remove dispensable weights, filters and other structures from neural networks. The goal of pruning is to reduce overall computational cost and memory footprint without inducing significant drop in performance of the network.

A common approach to mitigating performance drop after pruning is retraining: we continue to train the pruned models for some more epochs. In this paper, we are interested in approaches based on learning rate schedules to control the retraining. A well-known practice is *fine-tuning*, which aims to train the pruned model with a small fixed learning rate. More advanced learning rate schedules exist, which we generally refer to as *retraining*. The retraining step is a critical part in implementing network pruning, but it has been largely overlooked and tend to vary in each implementation including differences in learning rate schedules, retraining budget, hyperparameter choices, etc.

Recently, Renda et al. (2020) proposed a state-of-the-art technique for retraining pruned networks namely *learning rate rewinding* (LRW). Specifically, instead of fine-tuning the pruned networks with a fixed learning rate, usually the last learning rate from the original training schedule (Han et al., 2015; Liu et al., 2019), the authors suggested using the learning rate schedule from the previous $t$ epochs (i.e. rewinding). This seemingly subtle change in learning rate schedule led to an important result: LRW was shown to achieve comparable performance to more complex and computationally expensive

pruning algorithms while only utilizing simple norm-based pruning. Unfortunately, the authors did not provide the analysis to justify the improvement. In general, it is intriguing to understand the importance of a learning rate schedule and how it affects the final performance of a pruned model.

In this work, we study the behavior of pruned networks under different retraining settings. We found that the efficacy from retraining with learning rate rewinding is rooted in the use of a *large* learning rate, which helps pruned networks to converge faster after pruning. We demonstrate that the success of learning rate rewinding over fine-tuning is *not* exclusive to the learning rate schedule coupling with the original training process. Retraining with a large learning rate could outperform fine-tuning even with some modest retraining, e.g., for a few epochs, and regardless of network compression ratio.

We argue that retraining is of paramount importance to regain the performance in network pruning and should not be overlooked when comparing two pruning algorithms. This is evidenced by our extensive experiments: (1) *randomly* pruned network can outperform methodically pruned network with only (hyper-parameters free) modifications of the learning rate schedule in retraining, and (2) a simple baseline such as norm-based pruning can perform as well as as other complex pruning methods by using a large learning rate restarting retraining schedule.

The contributions of our work are as follows.

- We document a thorough experiment on learning rate schedule for the retraining step in network pruning with different pruning configurations;
- We show that learning rate matters: pruned models retrained with a *large* learning rate consistently outperform those trained by conventional fine-tuning regardless of specific learning rate schedules;
- We present a novel and counter-intuitive result achieved by solely applying large learning rate retraining: a **randomly** pruned network and a simple norm-based pruned network can perform as well as networks obtained from more sophisticated pruning algorithms.

Given the significant impact of learning rate schedule in network pruning, we advocate the following practices: learning rate schedule should be considered as a critical part of retraining when designing pruning algorithms. Rigorous ablation studies with different retraining settings should be made for a fair comparison of pruning algorithms. To facilitate reproducibility, we would release our implementation upon publication.

## 2 PRELIMINARY AND METHODOLOGY

Pruning is a common method to produce compact and high performance neural networks from their original large and cumbersome counterparts.We can categorize pruning approaches into **three** classes: *Pruning after training* - which consists of three steps: training the original network to convergence, prune redundant weights based on some criteria, and retrain the pruned model to regain the performance loss due to pruning (Li et al., 2016; Han et al., 2015; Luo et al., 2017; Ye et al., 2018; Wen et al., 2016; He et al., 2017); *Pruning during training* - we update the "pruning mask" while training the network from scratch, thus, allowing pruned neurons to be recovered (Zhu & Gupta, 2017; Kusupati et al., 2020; Wortsman et al., 2019; Lin et al., 2020b; He et al., 2019; 2018); *Pruning before training* - Inspired by the Lottery Ticket Hypothesis (Frankle & Carbin, 2019), some recent works try to find the sparsity mask at initialization and train the pruned network from scratch without changing the mask (Lee et al., 2019; Tanaka et al., 2020; Wang et al., 2020).

In this work, we are mainly concerned with the first category i.e. pruning after training which has the largest body of work to our knowledge. Traditionally, the last step is referred to as *fine-tuning*, i.e., continue to train the pruned model with a small learning rate obtained from the last epoch of the original model. This seemly subtle step is often overlooked when designing pruning algorithms.

Particularly, we found that the implementation of previous pruning algorithms have many notable differences in their retraining step: some employed a small value of learning rate (e.g. 0.001 on ImageNet) to fine-tune the network (Molchanov et al., 2016; Li et al., 2016; Han et al., 2015) for a small number of epochs, e.g., 20 epochs in the work by Li et al. (2016); some used a larger value of learning rate (0.01) with much longer retraining budgets, e.g., 60, 100 and 120 epochs respectively on ImageNet (Zhuang et al., 2018; Gao et al., 2020; Li et al., 2020); You et al. (2019); Li et al. (2020) respectively utilized 1-cycle (Smith & Topin, 2019) and cosine annealing learning rate schedule

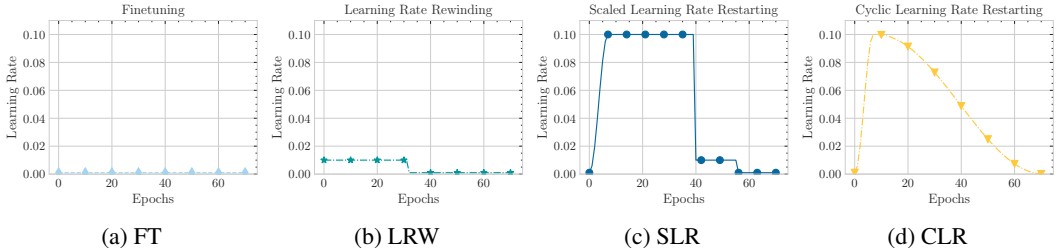

Figure 1: Learning rate with different schedules on CIFAR when retraining for 72 epochs. In (a), the learning rate is fixed to the last learning rate of original training (i.e. $0.001$). In (b), the learning rate is "rewound" to previous 72 epochs (which is $0.01$), and is dropped to $0.001$ after 32 epochs. In (c), after warming up the learning rate, we drop its value by the factor of $10\times$ at $50\%$ and $75\%$ of remaining epochs. In (d), we warm up the learning rate from the lowest to the highest value (of standard training) for the first few epochs, then decay the learning rate according to cosine function.

instead of conventional step-wise schedule. Despite such difference, the success of each pruning algorithm is only attributed to the pruning algorithm itself. This motivates us to ask the question: do details like learning rate schedule used for retraining matter?

In this section, we strive to understand the behavior of pruned models under different retraining configurations and how they impact the final performance. Specifically, we conduct experiments with different retraining schedules on simple baselines such as $\ell_1$-norm filters pruning (Li et al., 2016) and magnitude-based weights pruning (Han et al., 2015). We show that the efficacy of several pruning algorithms can be boosted by simply modifying the learning rate schedule. More importantly, the performance gain by retraining can be remarkable: the accuracy loss can drop to zero and in some cases better accuracy than baseline models can be achieved.

To analyze the effect of retraining a pruned network, we based on learning rate rewinding (Renda et al., 2020) and experiment with different retraining settings. Although in the previous work, Renda et al. (2020) demonstrated the efficacy of learning rate rewinding across datasets and pruning criteria, there is a lack of understanding of the actual reason behind the success of this technique. Here we hypothesize that the initial pruned network is a suboptimal solution, staying in a local minima. Learning rate rewinding succeeds because it uses a large learning rate to encourage the pruned networks to converge to another, supposedly better, local minima. Our experiment setups are as follows.

**Retraining techniques.** To verify this conjecture empirically, we conduct experiments with different learning rate schedules including learning rate rewinding (Renda et al., 2020) while varying pruning algorithms, network architectures and datasets. In this work, we consider the following retraining techniques:

1. FINE-TUNING (FT) Fine-tuning is the most common retraining techniques (Han et al., 2015; Li et al., 2016; Liu et al., 2019). In this approach, we continue train the pruned networks for $t$ epochs with the last (smallest) learning rate of original training.

2. LEARNING RATE REWINDING (LRW) Renda et al. (2020) propose to reuse the learning rate schedule of the original training when retraining pruned networks. Specifically, when retraining for $t$ epochs, we reuse the learning rate schedule from the previous $t$ epochs, i.e., rewinding.

3. SCALED LEARNING RATE RESTARTING (SLR): In this approach, we employ the learning rate schedule that is proportionally identical to the standard training. For example, the learning rate is dropped by a factor of $10\times$ at $50\%$ and $75\%$ of retraining epochs on CIFAR, which is akin to original training learning rate adjustment. The original learning rate schedule can be found in Appendix A.

4. CYCLIC LEARNING RATE RESTARTING (CLR): Instead of using stepwise learning rate schedule as *scaled learning rate restarting*, we leverage the 1-cycle (Smith & Topin, 2019), which is shown to give faster convergence speed than conventional approaches.

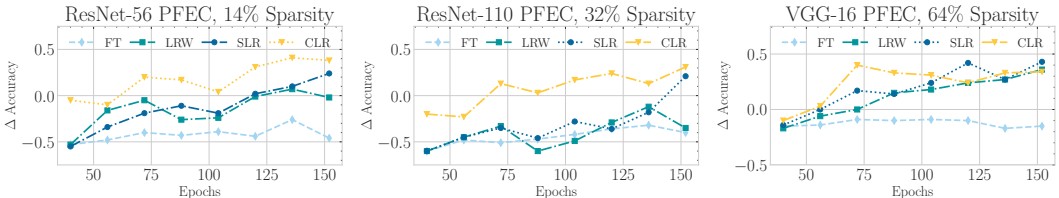

Figure 2: *One-shot structured* pruning on CIFAR-10 dataset using $\ell_1$-norm filters pruning (Li et al., 2016) while varying retraining budgets. As can be seen, learning rate schedule matters. Schedules that employ large learning rates (LRW, SLR, CLR) are significantly better than fine-tuning. Among them, CLR performs best in most cases.

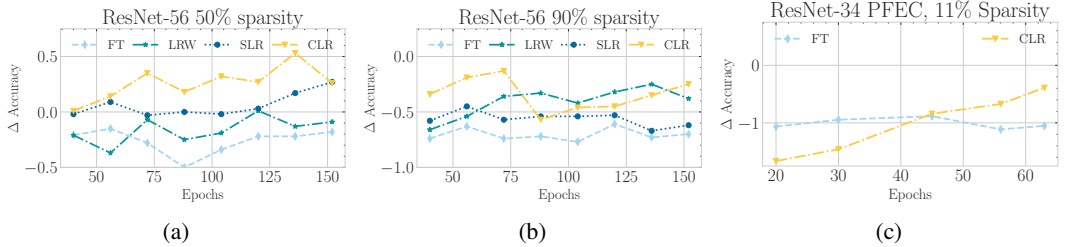

Figure 3: *One-shot unstructured* pruning on CIFAR-10 dataset using MWP (Han et al., 2015) ((a) and (b)) and *structured* pruning on ImageNet with $\ell_1$-norm filters pruning (Li et al., 2016) (c).

Note that for the last two strategies (SLR and CLR), we *warmup* the learning rate for $10\%$ of total retraining budget. For simplicity, we always use the *largest* learning rate of the original training for *learning rate restarting*. Specifically, the learning rate is increased from the smallest learning rate of original training to the largest one according to *cosine* function. The detailed learning rate schedule of each technique is depicted in Figure 1. See also Appendix F for the choice of warmup epochs.

**Pruning algorithms** We consider the following dimensions in our experiments. For pruning methods, we use $\ell_1$-norm filters pruning (PFEC) (Li et al., 2016) and (global) magnitude-based weights pruning (MWP) (Han et al., 2015) and evaluate them on the CIFAR-10, CIFAR-100 and ImageNet dataset. We examine both variations of pruning namely one-shot pruning and iterative pruning when comparing the proposed retraining techniques. Furthermore, we also experiment the CLR schedule on HRank (Lin et al., 2020a), Taylor Pruning (TP) (Molchanov et al., 2019) and Soft Filter Pruning (SFP) (He et al., 2018).

Our implementation and hyperparameters of $\ell_1$-norm filters pruning and magnitude weight pruning are based on the public implementation of Liu et al. (2019), which is shown to obtain comparable results with the original works. For remaining algorithms, we use official implementations with hyperparameters specified according to their papers. The detailed configurations of training and fine-tuning is provided in the Appendix B for interested readers.

**Evaluation.** For CIFAR-10 and CIFAR-100, we run each experiment three times and report "mean $\pm$ std". For ImageNet, we run each experiment once. These settings are kept consistently across architectures, pruning algorithms, retraining techniques, and ablation studies unless otherwise stated.

# 3    A CASE STUDY ON RETRAINING IN NETWORK PRUNING

## 3.1    RETRAINING COST AND PERFORMANCE TRADE-OFF

We first investigate the performance of retraining techniques while varying the retraining budget. Figure 2 illustrates the results of $\ell_1$-norm filters pruning (PFEC) (Li et al., 2016) with different retraining techniques on CIFAR-10. We can observe that, larger values of learning rate always attain higher performance than *fine-tuning* regardless of number of retraining epochs. Furthermore, with

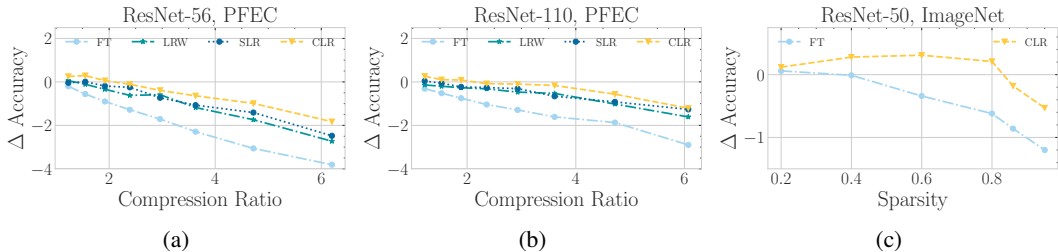

Figure 4: *Iterative* pruning on CIFAR-10 dataset using $\ell_1$-norm filters pruning Li et al. (2016) ((a) and (b)) and on ImageNet using magnitude-based weights pruning (Han et al., 2015) ((c)).

some additional retraining budgets, e.g., 80 to 120 epochs compare to 40 epochs in original works, we can attain much higher compression ratio with almost no accuracy drop (see Figure 5 in Appendix).

Figure 3 reports the accuracy of ResNet-56 pruned with magnitude-based weights pruning (Han et al., 2015) in a one-shot manner. We study the impact of different retraining schedules on low and high sparsity i.e. 50% and 90% respectively. It can be observed that under both setting learning rate restarting consistently outperforms fine-tuning across number of retraining epochs.

To verify the effectiveness of CLR on large-scale datasets, we conduct experiment using $\ell_1$-norm filters pruning with ResNet-34 on ImageNet in Figure 3(c). We find that CLR achieve lower performance than fine-tuning for low epochs. This is expected phenomenon since (Huang et al., 2017) also found that learning rate restarting on ImageNet usually requires training for 45 epochs while it only needs 40 epochs on CIFAR-10. When retraining with number of epochs higher than 45, CLR can reach higher accuracy than fine-tuning.

Thus, the value of LRW can be a good heuristic to choose the learning rate for retraining after pruning. A concrete example is that we achieve significant gains with Taylor pruning when retraining for only 25 epochs as shown in Table 8.

## 3.2 MODEL SIZE AND PERFORMANCE TRADE-OFF

In this section, we investigate the trade-off of between model size and performance of compact models under a fixed retraining budgets. Renda et al. (2020) found that learning rate rewinding usually saturate at half of original training, thus, we perform on retraining for 80 epochs on CIFAR-10 and 45 epochs on ImageNet.

We first experiment with iterative $\ell_1$-norm filters pruning on CIFAR-10 and report the results in Figure 4(a,b), we can observe that SLR and CLR also perform comparable or better than LRW in this setting. We then conduct experiments with iterative unstructured pruning on ImageNet with ResNet-50 to verify the superior of CLR compared to fine-tuning in Figure 4(c), we find that MWP with CLR significantly outperform fine-tuning and even increase the accuracy of the network until 80% sparsity (of convolutional layers).

In our experiments, CLR usually reaches slightly higher accuracy than LRW and SLR. A possible explanation is that 1-cycle schedule gives better convergence speed than conventional schedules (Smith & Topin, 2019), thus, bringing much better results when employing "low" retraining budgets.

## 3.3 EVALUATION WITH OTHER PRUNING ALGORITHMS

We investigate whether the effectiveness of learning rate restarting holds with other pruning algorithms than simple norm-based pruning. As the performance of larger learning rate schedules such as LRW, SLR, and CLR are rather similar, we select CLR for use in this experiment. Particularly, we examine *CLR* with SFP (He et al., 2018) in this section..

**Soft Filters Pruning** (He et al., 2018). Previous sections demonstrate that the success of large learning rate hold for a wide range of pruning algorithms, network architectures and datasets. However, we note that all examined pruning algorithms followed the *pruning-retraining* paradigm, i.e., it is necessary to retrain the pruned model. In this part, we investigate *regularizing* where we

Table 1: Results (accuracy) for soft filters pruning (He et al., 2018). The "w/o FT" column demonstrate the performance of networks follow original work (do not retrain after final prune). "FT" column indicates results of networks when fine-tuning for 200 more epochs. "CLR-x" columns show accuracy of networks after retraining with CLR for $x$ more epochs. "%PF" indicates number of pruned filters.

| Model | %PF | w/o FT | FT | CLR-50 | CLR-100 | CLR-200 |
|---|---|---|---|---|---|---|
| CIFAR-10 | 30 | 92.74±0.28 | 92.85±0.21 | 92.57±0.19 | 93.00±0.02 | **93.18 ± 0.25** |
| ResNet-56 | 40 | 91.95±0.29 | 91.94±0.26 | 92.19±0.16 | 92.23±0.12 | **92.78 ± 0.34** |
| CIFAR-10 ResNet-110 | 40 | 92.25±0.55 | 92.23±0.50 | 92.30±0.37 | 92.79±0.38 | **92.91 ± 0.41** |
| CIFAR-100 | 30 | 68.92 ±0.33 | 69.41±0.38 | 70.27±0.40 | 70.24±0.43 | **70.29 ± 0.46** |
| ResNet-56 | 40 | 67.69±0.41 | 67.64±0.50 | 68.78±0.23 | **69.12 ± 0.22** | 68.88±0.44 |
| CIFAR-100 | 30 | 70.93±0.58 | 70.94±0.58 | 71.77±0.60 | 71.74±0.63 | **72.00 ± 0.21** |
| ResNet-110 | 40 | 68.41±0.26 | 69.04±0.41 | 70.22±0.14 | 70.05±0.27 | **70.34 ± 0.24** |

gradually, softly prune the network during the course of training. In contrast to *pruning-retraining*, the latter approach does not require further training after final pruning. That being said, we still make a case study to verify if retraining does improve the performance of models pruned with this type of algorithms.

To make a fair comparison between fine-tuning and no fine-tuning, we randomly split the conventional training set of CIFAR-10/CIFAR-100 (including 50000 images) to *train* (90% images of total set) and *val* (10% remaining images) set and we then report the result of best-validation models on "standard" *test* set (including 5000 images).

We report the performance of pruned network without fine-tuning, with fine-tuning and with learning rate restarting (CLR) in Table 1. We find that applying fine-tuning on these model usually give very fast convergences speed (roughly in first 20 epochs). However, fine-tuned models exhibit negligible gain then pruned models without fine-tuning. In contrast, CLR significantly improves the performance even with small retraining budgets. Interested readers can find our results with other pruning algorithms namely HRank (Lin et al., 2020a) and Taylor pruning (Molchanov et al., 2019) in the Appendix C.

## 3.4 SCALING ORIGINAL TRAINING

Liu et al. (2019) present empirical evidence that 3-stage pipeline of *structured* pruning is inferior to training compact models from scratch if training budgets of compact models are scaled proportion to FLOPs ratio between original and pruned model. Following Evci et al. (2019); Gale et al. (2019), we conduct experiments to compare this strong baseline with $\ell_1$-norm filters pruning while employing CLR on CIFAR-10, CIFAR-100 and ImageNet. Specifically, we compare pruned networks retrained with fine-tuning and CLR mechanism with Scratch-B(udget) and Scratch-E(pochs) which train pruned networks from scratch with the same computational budget and epochs respectively [1].

When evaluating with CIFAR-10 and CIFAR-100, we found that the validation accuracies of Scratch-B, Scratch-E, and CLR are very similar to each other, and therefore, the comparisons and ranking could be very sensitive to the variance of the accuracy. To reduce variance, we employ the following training scheme. First, we re-run each experiment 5 times on CIFAR-10 and CIFAR-100 (more than previous experiments to reduce noise). Second, we randomly (and independently between each run) split the original training set of CIFAR-10 to *training* (90% images of total set) and *validation* set (10% images of total set) and report the performance of *best validation* models on original *test set*. Note that for ImageNet, we re-run the experiments 3 times. We found that for ImageNet, the variance is negligible and thus we simply report the *best validation* accuracy.

For retraining, we adopt the same retraining budgets as in (Liu et al., 2019; Li et al., 2016) (40 and 20 epochs for CIFAR-10 and ImageNet respectively). Since the total number of retraining epochs is

---

[1] For Scratch-E, we use 160 (original training) + 40 (retraining) = 200 epochs while training from scratch instead of 160 as in (Liu et al., 2019)

Table 2: Results (accuracy) for $\ell_1$-norm based filter pruning (Li et al., 2016). Configurations of Model and Pruned Model are both from the original paper (Li et al., 2016). The results of "Scratch-E" and "Scratch-B" on ImageNet are taken directly from work of Liu et al. (2019). Top- and second-ranked results are highlighted in **bold blue** and blue.

| Dataset | Model | Unpruned | Version | Fine-tuning | Scratch-E | Scratch-B | CLR |
|---------|-------|----------|---------|-------------|-----------|-----------|-----|
| CIFAR-10 | VGG-16 | $93.01 \pm 0.21$ | - | $93.09 \pm 0.19$ | $93.01 \pm 0.27$ | $\mathbf{93.24 \pm 0.25}$ | $93.12 \pm 0.26$ |
| | ResNet-56 | $92.32 \pm 0.32$ | A | $92.18 \pm 0.28$ | $92.60 \pm 0.17$ | $\mathbf{92.77 \pm 0.19}$ | $92.71 \pm 0.33$ |
| | | | B | $92.06 \pm 0.23$ | $92.39 \pm 0.22$ | $92.31 \pm 0.23$ | $\mathbf{92.41 \pm 0.19}$ |
| | ResNet-110 | $92.91 \pm 0.38$ | A | $92.81 \pm 0.39$ | $92.97 \pm 0.22$ | $93.04 \pm 0.18$ | $\mathbf{93.08 \pm 0.38}$ |
| | | | B | $92.34 \pm 0.33$ | $92.63 \pm 0.70$ | $\mathbf{93.20 \pm 0.25}$ | $93.03 \pm 0.31$ |
| CIFAR-100 | VGG-16 | $71.26 \pm 0.23$ | - | $69.97 \pm 0.32$ | $\mathbf{71.37 \pm 0.07}$ | $71.72 \pm 0.34$ | $71.18 \pm 0.07$ |
| | ResNet-56 | $69.51 \pm 0.25$ | A | $69.55 \pm 0.54$ | $\mathbf{70.15 \pm 0.52}$ | $69.74 \pm 0.57$ | $70.04 \pm 0.30$ |
| | | | B | $69.53 \pm 0.28$ | $69.64 \pm 0.24$ | $69.91 \pm 0.53$ | $\mathbf{70.06 \pm 0.35}$ |
| | ResNet-110 | $70.59 \pm 0.36$ | A | $70.71 \pm 0.16$ | $71.11 \pm 0.05$ | $70.61 \pm 0.25$ | $\mathbf{71.29 \pm 0.25}$ |
| | | | B | $69.54 \pm 0.27$ | $70.61 \pm 0.45$ | $\mathbf{70.79 \pm 0.22}$ | $70.52 \pm 0.31$ |
| ImageNet | ResNet-34 | 73.30 | A | $72.96 \pm 0.06$ | 72.56 | 73.03 | $\mathbf{73.44 \pm 0.06}$ |
| | | | B | $72.50 \pm 0.08$ | 72.29 | 72.91 | $\mathbf{73.14 \pm 0.03}$ |

limited, for ImageNet, we restart the learning rate to $0.01$ (instead of $0.1$ in the previous section) to guarantee the convergence of the models.

The results of pruned network via PFEC are shown in Table 2. We can observe that for CIFAR-10, the results of fine-tuning and Scratch-B/E are aligned with the prior work of Liu et al. (2019) while CLR achieve slightly better results than Scratch-E and comparable with Scratch-B. However, PFEC+CLR significantly outperform both fine-tuning and Scratch-B by a large margin.

**Discussion.** Our empirical results suggest that practitioners should employ retraining with large learning rate schedules as an alternative technique for fine-tuning to obtain compact models with better performance after network pruning. In our results, cyclic learning rate restarting (CLR) is slightly more efficient than scaled learning rate restarting (SLR) and learning rate rewinding (LRW).

## 4 INTERPLAY OF PRUNING ALGORITHMS AND RETRAINING SCHEMES

Section 3 suggests that learning rate schedule can significantly improve the performance of several pruning algorithms. We notice that there are notable differences between settings of implementation of network pruning especially in retraining phase. In this section, we show that the difference between implementation could easily lead to misleading results and unfair comparisons between pruning algorithms.

### 4.1 A STRONG BASELINE: $\ell_1$-NORM FILTERS PRUNING WITH CLR

In this section, we demonstrate that even with **same** retraining budgets, utilizing simple *CLR* with $\ell_1$-norm filters pruning can achive comparable or exceed the performance of more sophisticated saliency metrics *without* meticulous hyperparameters searching. The implementation details are as follows.

**Our baseline** For pruning, we adopt the pretrained of Torchvision and apply $\ell_1$-norm Filters Pruning on these models. In our implementation, the number of removed filters in each block are approximately equal so that the final pruned models have similar number of parameters with the compared one. Thus, if any, this setting should favor other approaches involving laborious sensitive analysis. For retraining, we apply the *CLR* learning rate schedule while choosing the maximum value of learning rate according to *learning rate rewinding*. For pruning algorithms that only retrain for small number of epochs (e.g. 25 epochs in case of Taylor Pruning) we also *restart* the learning rate value to the slightly higher value of $0.01$.

**Generative Adversarial Learning (GAL)** Lin et al. (2019) suggest to leverage generative adversarial learning, which learns a sparse soft mask in a label-free and an end-to-end manner, to effectively

Table 3: Comparing the performance of pruned network via PFEC + CLR and GAL on ImageNet. The results of GAL are taken directly from original papers.

| Model | Unpruned Top-1 | Method | Param ↓ % | FLOPs ↓ % | Top-1 |
|-------|----------------|--------|-----------|-----------|-------|
| ResNet-50 | 76.15 | GAL-0.5 | 17.2 | 43.0 | 71.95 |
| | | PFEC + CLR | 17.6 | - | **75.26** |
| | | GAL-1 | 42.6 | 61.4 | 69.88 |
| | | PFEC + CLR | 43.1 | - | **73.81** |

Table 4: Comparing the performance of pruned network via PFEC + CLR and HRankPlus on ImageNet. The results of HRankPlus are taken directly from official Github repository.

| Model | Unpruned Top-1 | Method | Param ↓ % | FLOPs ↓ % | Top-1 |
|-------|----------------|--------|-----------|-----------|-------|
| ResNet-50 | 76.15 | HRankPlus | 40.8 | 44.8 | 75.56 |
| | | PFEC + CLR | 41.4 | - | **75.59** |
| | | HRankPlus | 56.7 | 62.8 | 74.19 |
| | | PFEC + CLR | 56.9 | - | **74.39** |

solve the optimization problem. The original work retrain the pruned network for 30 epochs with batch size of 32 and initial learning rate of 0.01 (which is decayed by factor of 10 every 10 epochs). Thus, we use *CLR* with learning rate of 0.01 and batch size of 32 in our experiments when comparing with *GAL*. Table 3 shows the results of PFEC + CLR when retrained under same settings with GAL. We can see that PFEC+CLR can create a very strong baseline compare with GAL.

**HRankPlus** An extension of HRank (Lin et al., 2020a) which is better than HRank in both retraining efficiency and performance [2]. Inspired by the discovery that average rank of feature maps of a filter is consistent, the authors suggested to iteratively remove low-rank feature maps that contain less information. In the original implementation, the authors retrain the pruned networks for 90 epochs with initial learning rate of 0.1 and also utilize label smoothing. For simplicity, we do not use label smoothing in our implementation. As shown in Table 12, PFEC+CLR also attain similar results with HRankPlus approach.

Additional results on CIFAR-10 and Discrimination-aware Channel Pruning (Zhuang et al., 2018), Taylor Pruning (Molchanov et al., 2019), Provable Filters Pruning (Liebenwein et al., 2020) are reported in Appendix D for interested readers.

## 4.2 RANDOM PRUNING WITH LEARNING RATE RESTARTING

In last subsection, we demonstrated that slight modification in retraining of simple $\ell_1$-norm Filters Pruning can exceed the perfomance of much more sophisticated pruning algorithms. In this subsection, we investigate the interplay between pruning saliency metrics and retraining configurations by comparing accuracy of randomly pruned networks with the *original performance* of methodically pruned networks.

We directly adopt the original implementation from the authors of Taylor Pruning, HRankPlus for comparisons. For PFEC and MWP we make use of the implementation of Liu et al. (2019). We selected these works because they do not utilize a similar learning rate value as LRW in the original implementation, i.e., the authors applied a smaller value then the heuristic of LRW. The detailed training recipe of the original works can be found in Appendix E.1. To make the randomly pruned network have the same structure as the methodically pruned networks, i.e., equal number of filters per layer, we propose to replace the importance score of each neuron estimated by these methods by a uniformly random score value.

For retraining randomly pruned networks, we adopt the same principle at described in Section 4.1. Table 5 presents novel results of performance of random pruning + CLR with corresponding results of considered pruning algorithms. It is worth pointing out that we report the *original* results of all

---

[2]https://github.com/lmbxmu/HRankPlus

Table 5: Results of networks when applying random pruning and methodically pruning algorithms. "Original" column presents accuracy of pruned network reported in original papers. "R-CLR" presents the results of Random Pruning with CLR.

| Dataset | Method | Model | Params $\downarrow$ % | FLOPs $\downarrow$ % | Original | R-CLR |
|---------|--------|-------|--------|--------|----------|-------|
| CIFAR-10 | HRankPlus | ResNet-56 | 70.0 | 74.1 | 92.32 | **92.40 $\pm$ 0.16** |
| | | ResNet-110 | 68.3 | 71.6 | 93.23 | **93.37 $\pm$ 0.04** |
| | | DenseNet-40 | 61.9 | 59.9 | 93.66 | **93.71 $\pm$ 0.05** |
| | | VGG-16 | 87.3 | 78.6 | **93.10** | 93.06 $\pm$ 0.07 |
| ImageNet | PFEC | ResNet-34 A | 2.3 | 15.9 | 72.96 $\pm$ 0.06 | **73.47 $\pm$ 0.08** |
| | | ResNet-34 B | 10.8 | 24.2 | 72.50 $\pm$ 0.08 | **73.05 $\pm$ 0.05** |
| | Taylor Pruning | ResNet-50 72% | 44.5 | 45.0 | 74.50 | **74.91** |
| | | ResNet-50 81% | 30.1 | 35.0 | 75.48 | **75.54** |
| | | ResNet-50 91% | 11.4 | 20.0 | **76.43** | 75.93 |

methodical pruning are taken directly from their papers respectively. In our experiments, Random Pruning + CLR could surpass the performance of sophisticated saliency metrics even on large-scale and challenging dataset such as ImageNet with only minimal change in learning rate schedule and initial learning rate value.

Comprehensive results of random pruning with different compression ratio for $\ell_1$-norm Filters Pruning and MWP is reported in Appendix E.2.

These results suggest that retraining techniques, e.g., learning rate restarting and learning rate schedule, play a pivotal role to final performance. Thus, in order to perform fair comparison of different methods, one should be cautious of this seemingly subtle detail. While it is unclear to set up a fair comparison between pruning algorithms that belong to different categories, e.g., pruning before/during/after training, iterative vs. oneshot pruning, we advocate standardizing (a set of) retraining configurations for each algorithm group or thorough hyperparamters searching to find the best configuration of each pruning method when evaluating different algorithms.

## 5 Discussion and Conclusion

In this work, we conducted extensive experiments to show that learning rates do matter in achieving good performance of pruned neural networks. We concluded that compared to traditional fine-tuning, learning rate restarting in general is an efficient way to retrain pruned networks to recover performance drop due to pruning. The success of learning rate rewinding is accounted by the use of large learning rates in the retraining. We believe that these findings help raise awareness of proper use of learning rate schedule when desiging pruning algorithms, standardizing empirical experiments and allowing fair comparisons. Our takeaway message is:

*Pruning algorithms should be compared in the same retraining configurations.*

Our work is not without limitations. So far we investigated with hyperparameters identical to those of the original implementations of the pruning algorithms, and only experiment with different learning rate schedules. We also limited our experiments to the image classification task in computer vision. Considering more datasets and other domains is a great research avenue for future work.

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

# APPENDIX

## A   NETWORK PRUNING FORMULATION

Let us start with a formal definition of the network pruning problem. Let $\boldsymbol{\theta} \in \mathbb{R}^D$ be the parameters of a (large) neural network that needs to be pruned, $\mathcal{D}$ be the dataset for training. The parameters $\boldsymbol{\theta}$ is updated to minimize the loss function $\mathcal{L}(\boldsymbol{\theta}; \mathcal{D})$. Define *sparsity mask* $\mathbf{m} \in \{0, 1\}^D$ indicating if a weight should be kept or removed. Similar to Molchanov et al. (2016), we can formulate the pruning algorithms as finding the optimal sparsity mask such that:

$$\mathbf{m}^* = \underset{\mathbf{m}}{\operatorname{argmin}} \; \mathcal{L}(\boldsymbol{\theta}^* \odot \mathbf{m}; \mathcal{D}) - \mathcal{L}(\boldsymbol{\theta}^*; \mathcal{D}) \tag{1}$$

where $\odot$ denotes the Hadamard (element-wise) product, $\boldsymbol{\theta}^*$ represents the (locally) optimal solution of $\boldsymbol{\theta}$. An alternative approach is minimizing the different between final output of two networks (Lin et al., 2019):

$$\mathbf{m}^* = \underset{\mathbf{m}}{\operatorname{argmin}} \; \|\mathcal{F}(\boldsymbol{\theta}^* \odot \mathbf{m}; \mathcal{D}) - \mathcal{F}(\boldsymbol{\theta}^*; \mathcal{D})\|_2^2 \tag{2}$$

where $\mathcal{F} : \mathbb{R}^D \to \mathbb{R}^{D'}$ is the function mapping input feature (i.e. images) to the final output (before softmax).

Theoretically speaking, as the pruning formulation only takes the loss function into account, and there is a caveat that pruning can result in a set of weights that belong to a bad local minima, making the network generalize poorly. In practice, it is common to observe such issues from the performance drop of the pruned model. In such cases, a fine-tuning process can be applied to further optimize the weights, mitigating the accuracy drop. However, as what fine-tuning means, the weights might be only slightly adjusted without any guarantee about the final network performance.

## B   TRAINING CONFIGURATION

The implementation of $\ell_1$-norm filters pruning (PFEC) Li et al. (2016) and magnitude-based weights pruning (MWP) Han et al. (2015) are adopted from the work of Liu et al. (2019). The implementations of other algorithms are taken from the authors' official repositories.

For simplicity, we adopt Pytorch's pretrained models for ImageNet. The unpruned models (used for $\ell_1$-norm filters pruning) are trained with below configurations.

Table 6: Training configuration for unpruned models. To train CIFAR-10, we use Nesterov SGD with $\beta = 0.9$, batch size 64, weight decay 0.0001 for 160 epochs. To train ImageNet, we use Nesterov SGD with $\beta = 0.9$, batch size 32, weight decay 0.0001 for 90 epochs.

| Dataset | Network | #Params. | Learning rate (t = training epoch) | Test accuracy |
|---------|---------|----------|-------------------------------------|---------------|
| CIFAR-10 | ResNet-56 | 0.85M | $\alpha = \begin{cases} 0.1 & t \in [0, 80) \\ 0.01 & t \in [80, 120) \\ 0.001 & t \in [120, 160) \end{cases}$ | $93.46 \pm 0.21\%$ |
|  | ResNet-110 | 1.73M |  |  |
| ImageNet | ResNet-18 | 11.69M | $\alpha = \begin{cases} 0.1 & t \in [0, 30) \\ 0.01 & t \in [30, 60) \\ 0.001 & t \in [60, 90) \end{cases}$ | 69.76% top-1 |
|  | ResNet-34 | 21.8M |  | 73.30% top-1 |
|  | ResNet-50 | 25.5M |  | 76.15% top-1 |

### B.1   ADDITIONAL RESULTS

In addition results in Figure 2, we conduct experiments to examine impact of different retraining techniques to networks pruned with low and high compression ratios. The results are shown in Figure 5.

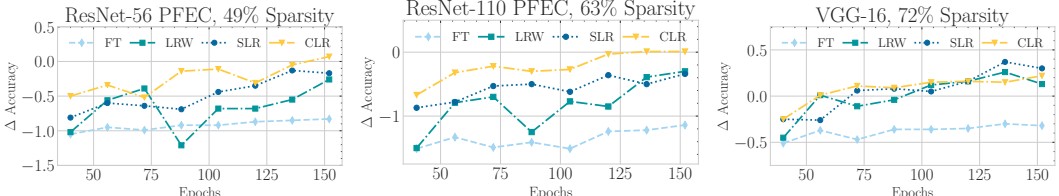

Figure 5: Results from pruning with high compression ratios for *one-shot structured* pruning on CIFAR-10 dataset using $\ell_1$-norm filters pruning (Li et al., 2016). With a proper learning rate schedule, it is possible to achieve almost no accuracy drop while having much more compact models and using slightly more training budgets.

Table 7: Results (accuracy) for HRank ilters pruning (Lin et al., 2020a) on CIFAR-10. "Pruned + FT†" is the model pruned from the large model with *original* fine-tuning scheme that are reported. "Prune + FT" and "Prune + CLR" are results of our runs with original retraining and CLR respectively. Configurations of Model and Pruned Model are both from the original paper.

| Model | Unpruned | Prune + FT † | Prune + FT | Prune + CLR | FLOPs ↓ % | Params ↓ % |
|-------|----------|--------------|------------|-------------|-----------|------------|
| VGG-16 (BN) | 93.93 | 92.34 | 91.97 | **92.53** | 65.3% | 82.10% |
| ResNet-56 | 93.26 | 93.17 | 92.97 | **93.16** | 50.0% | 42.4% |
| ResNet-110 | 93.50 | 94.23 | 93.84 | **93.90** | 37.9% | 38.7% |
| DenseNet-40 | 94.81 | 94.24 | 94.10 | 94.10 | 40.8% | 30.5% |

## C    EVALUATING CLR WITH OTHER PRUNING ALGORITHMS

**Taylor Pruning** (Molchanov et al., 2019). In this method, the authors estimated the contributions of each filter to the final performance by utilizing the first- and second-order Taylor expansion and performing iterative pruning. In the original implementation, the authors fine tuned the network with a small learning rate (0.001) for 25 epochs on ImageNet. In our implementation, we opt to evaluate the accuracy of network retrained with learning rate restarting (CLR) with a relatively small budget. Therefore, we keep the retraining budget the same and only increase the learning rate to 0.01.

Table 8 shows the accuracy of ResNet-50 on ImageNet when pruning 72% of network weights. It can be seen that for a small retraining budget, employing learning rate restarting also gives significant gain in performance.

**HRank** (Lin et al., 2020a). Inspired by the discovery that average rank of feature maps of a filter is consistent, the authors suggested to iteratively remove low-rank feature maps that contain less information. Particularly, we iteratively remove filters in each layer and then retrain the network for 30 epochs each time. We examine the performance of pruned models with HRank on CIFAR while employing CLR at each fine-tuning step. Due to the heavily expensive computational cost for retraining, we only run each model once for HRank. In the original version of this method, the authors used the learning rate of 0.01 and drop by $10\times$ at epochs 5 and 10. In our implementation, we restart the learning rate to 0.1 and apply the CLR schedule.

The quantitative results are shown in Table 7. In our experiments, CLR also attains comparable or better results than proposed retraining methods in original work.

## D    STRONG BASELINES WITH $\ell_1$-NORM FILTERS PRUNING

### D.1    CIFAR-10

Furthermore, we show that by naively neglecting retraining configurations (i.e. budget), we can also reach state-of-the-art results on CIFAR-10 with PFEC Li et al. (2016). Specifically, we prune two "standard" models namely ResNet-56 and ResNet-110 and retrain with CLR for 152 epochs, and then we compare with more sophisticated algorithms in Table 9.

Table 8: Results (accuracy) for Taylor filters pruning (Molchanov et al., 2019) on ImageNet. All columns have same meaning with corresponding columns in Table 7.

| Model | Unpruned | Prune + FT | Prune + CLR | FLOPs ↓ % | Params ↓ % |
|---|---|---|---|---|---|
| Taylor-FO-BN-56% | | 71.69 | **72.51** | 67.2 | 66.8 |
| Taylor-FO-BN-72% | 76.15 | 74.50 | **75.22** | 45.0 | 44.5 |
| Taylor-FO-BN-81% | | 75.48 | **75.67** | 35.0 | 30.1 |

Table 9: Results of ResNet-56 and ResNet-110 on CIFAR-10. The performance of other pruning algorithms are taken directly from original papers.

| Model | Method | Unpruned | Params ↓ | FLOPs ↓ | Prune | Acc. ↓ |
|---|---|---|---|---|---|---|
| | PFEC (Li et al., 2016) | 93.04 | 14.1 | 27.6 | 93.06 | -0.02 |
| | NISP (Yu et al., 2018) | 93.26 | 42.4 | 35.5 | 93.01 | 0.25 |
| ResNet-56 | HRank (Lin et al., 2020a) | 93.26 | 42.4 | 50.0 | 93.17 | 0.09 |
| | GAL (Lin et al., 2019) | 93.26 | - | 37.6 | 92.98 | 0.28 |
| | PFEC + CLR (ours) | 93.21 | 48.7 | - | 93.29 | -0.07 |
| | PFEC (Li et al., 2016) | 93.53 | 32.6 | 38.7 | 93.30 | -0.23 |
| ResNet-110 | GAL-0.5 (Lin et al., 2019) | 93.50 | 44.8 | 48.4 | 92.74 | 0.76 |
| | HRank (Lin et al., 2020a) | 93.50 | 59.2 | 58.2 | 93.36 | 0.14 |
| | PFEC+CLR (ours) | 93.56 | 64.2 | - | 93.69 | -0.13 |

## D.2 IMAGENET

**Taylor Pruning (Molchanov et al., 2019)**    In this method, the authors estimated the contributions of each filter to the final performance by utilizing the first- and second-order Taylor expansion and performing iterative pruning.

Original work retrains pruned networks for 25 epochs with learning rate of 0.001 with the exception for *Taylor-FO-BN-56%*. Table 10 demonstrates the performance of *PFEC+CLR* compare with original works of Molchanov et al. (2019). In most cases, *PFEC+CLR* attains higher or comparable accuracy with *Taylor Pruning*.

Table 10: Comparing the performance of pruned network via PFEC + CLR and Taylor Pruning on ImageNet. The results of Taylor Pruning are taken directly from original papers.

| **Model** | **Method** | **Param ↓ %** | **FLOPs ↓ %** | **Unpruned Top-1** | **Top-1** |
|---|---|---|---|---|---|
| ResNet-34 | Taylor-FO-BN-82% | 21.1 | 22.3 | 73.31 | 72.83 |
| | PFEC + CLR | 22.5 | - | 73.30 | **73.01** |
| | Taylor-FO-BN-56% | 66.8 | 67.2 | 76.18 | **71.69** |
| | PFEC + CLR | 68.8 | - | 76.15 | 70.70 |
| | Taylor-FO-BN-72% | 44.5 | 45.0 | 76.18 | 74.50 |
| ResNet-50 | PFEC + CLR | 44.5 | - | 76.15 | **75.04** |
| | Taylor-FO-BN-81% | 30.1 | 35.0 | 76.18 | 75.48 |
| | PFEC + CLR | 30.1 | - | 76.15 | **75.79** |
| | Taylor-FO-BN-91% | 11.4 | 20.0 | 76.18 | **76.43** |
| | PFEC + CLR | 12.5 | - | 76.15 | 76.37 |

**Discrimination-aware Channel Pruning (Zhuang et al., 2018)**    The authors introduce additional discrimination-aware losses into the network to increase the discriminative power of intermediate layers and then select the most discriminative channels for each layer by considering the additional loss and the reconstruction error.

We use the retraining budget of 60 epochs and set initial learning rate to 0.01 same as Zhuang et al. (2018). However, we employ the 1-cycle learning rate instead of stepwise learning rate as original work. The detailed comparison is presented in Table 11.

Table 11: Comparing the performance of pruned network via PFEC + CLR and DCP on ImageNet. The results of DCP are taken directly from original papers.

| Model | Param ↓ % | FLOPs ↓ % | Method | Unpruned Top-1 | Top-1 |
|---|---|---|---|---|---|
| | 28.1 | 27.1 | DCP | 69.64 | 69.21 |
| | 31.9 | - | PFEC + CLR | 69.76 | **69.31 ± 0.06** |
| ResNet-18 | 47.1 | 46.1 | DCP | 69.64 | 67.35 |
| | 50.6 | - | PFEC + CLR | 69.76 | **67.38** |
| | 65.7 | 64.1 | DCP | 69.64 | **64.12** |
| | - | - | PFEC + CLR | 69.76 | 64.08 |
| | 33.3 | 35.7 | DCP | 76.01 | **76.40** |
| | 33.7 | - | PFEC + CLR | 76.15 | 76.03 |
| ResNet-50 | 51.4 | 55.5 | DCP | 76.01 | 74.95 |
| | 51.5 | - | PFEC + CLR | 76.15 | **75.16** |
| | 65.9 | 71.1 | DCP | 76.01 | 72.75 |
| | 66.1 | - | PFEC + CLR | 76.15 | **72.92** |

**Provable Filters Pruning (Liebenwein et al., 2020)** This algorithm uses a small batch of input data points to assign a saliency score to each filter and constructs an importance sampling distribution where filters that highly affect the output are sampled with correspondingly high probability.

Original implementation of Liebenwein et al. (2020) retrain the pruned network on ImageNet for 90 epochs with standard learning rate schedule (drop learning rate by factor of 10 after 30 epochs) with batch size of 256 and weight decay of 0.0001. In our implementation, pruned networks are trained for 90 epochs with maximum learning rate of 0.1 while keeping the same configurations for all other hyperparameters as original work.

Table 12: Comparing the performance of pruned network via PFEC + CLR and Provable Filters Pruning (PFP) on ImageNet. The results of PFP are taken directly from original paper (Liebenwein et al., 2020).

| Model | Method | Param ↓ % | FLOPs ↓ % | Unpruned Top-1 | Top-1 |
|---|---|---|---|---|---|
| ResNet-50 | PFP (lowest top-1 err.) | 18.0 | 10.8 | 76.13 | 75.91 |
| | PFEC + CLR | 18.1 | - | 76.15 | **76.56** |
| | PFP (within 1.0% top-1) | 44.0 | 30.1 | 76.13 | 75.21 |
| | PFEC + CLR | 44.3 | - | 76.15 | **75.38** |

# E    RANDOM PRUNING WITH CLR

## E.1    RETRAINING CONFIGURATION

We compare the configuration between original implementation of aforementioned pruning methods and corresponding random pruning in Table 13. Note that for HRankPlus, the original implementation employs different values of weight decay for each model: $0.005, 0.006, 0.005, 0.002$ for VGG-16, ResNet-56, ResNet-110, DenseNet-40 respectively. We found that these value is relatively higher than "conventional" values (0.0001) making the models hard to converge after restarting. Thus, we use weight decay of 0.0005 through out experiments with HRankPlus. In all our experiments, other details such as batch size, retraining budget are set to similar value of original implementation.

Table 13: Training configurations of original pruning methods and random pruning.

| Method | Model | Original | | Random Pruning |
|---|---|---|---|---|
| HRankPlus | ResNet-56

ResNet-110

DenseNet-40 | $\alpha = \begin{cases} 0.01 & t \in [0, 150) \\ 0.001 & t \in [151, 225) \\ 0.0001 & t \in [226, 300) \end{cases}$ | | $\begin{cases} \alpha_{\text{init}} = 0.001 \\ \alpha_{\text{max}} = 0.1 \\ \alpha_{\text{min}} = 0.00001 \end{cases}$ |
| | VGG-16 | $\alpha = \begin{cases} 0.01 & t \in [0, 50) \\ 0.001 & t \in [51, 100) \\ 0.0001 & t \in [101, 150) \end{cases}$ | | |
| PFEC | ResNet-34 | $\alpha = 0.001$ | $t \in [0, 20)$ | $\begin{cases} \alpha_{\text{init}} = 0.0001 \\ \alpha_{\text{max}} = 0.01 \\ \alpha_{\text{min}} = 0.000001 \end{cases}$ |
| Taylor Pruning | ResNet-50 | $\alpha = 0.001$ | $t \in [0, 25)$ | |

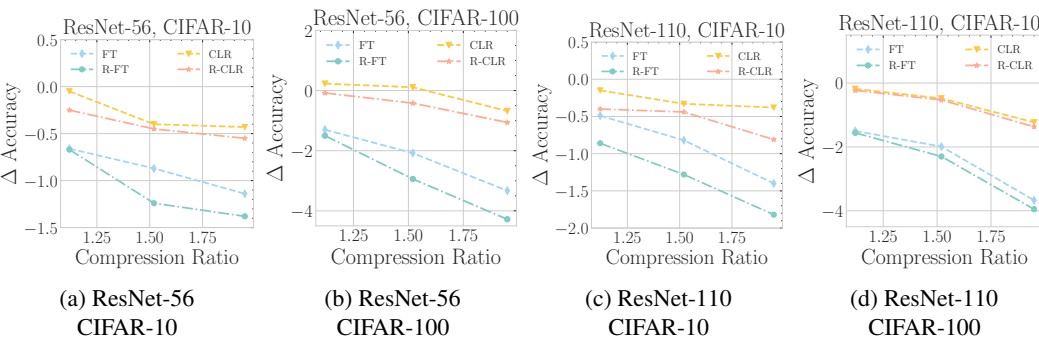

(a) ResNet-56 (b) ResNet-56 (c) ResNet-110 (d) ResNet-110
CIFAR-10  CIFAR-100  CIFAR-10  CIFAR-100

Figure 6: *One-shot structured* pruning on CIFAR-10 dataset using PFEC (Li et al., 2016) and randomly filters pruning with different retraining schemes.

## E.2 RANDOM PRUNING WITH VARIOUS COMPRESSION RATIO

$\ell_1$-**norm Filters Pruning** Figure 6 illustrates the performance of networks pruned with PFEC and random pruning on CIFAR-10 and CIFAR-100 when retraining with 40 epochs – the same setting used by Li et al. (2016). We can see that randomly pruned networks consistently achieve *superior* performance than methodically pruned networks (fine-tuned with standard learning rate schedule) in terms of accuracy. However, random pruning obtain lower accuracy than PFEC when using identical retraining techniques.

**Magnitude-based Weights Pruning** We extend the scope of experiment in Sec 4.2 to *unstructured* pruning and analyze performance of MWP with CLR. Figure 7 represents the results of random pruning with CLR and methodically pruning while varying difference compression ratio in both iterative and oneshot pruning manner. Specifically, we retrain the trimmed network for 40 epochs. We can observe that though Random Pruning + CLR can achieve higher accuracy with low sparsity, the performance of randomly pruned network immensely reduced with the increasing of compression ratio.

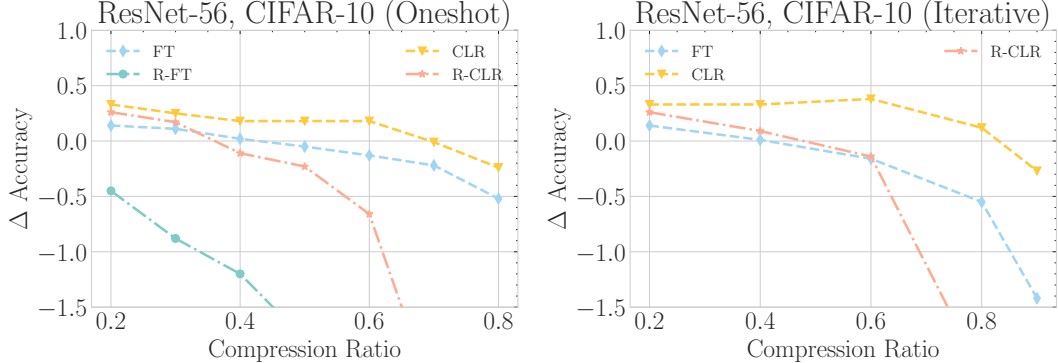

Figure 7: *One-shot* and *Iterative unstructured* pruning on CIFAR-10 dataset using MWP (Han et al., 2015) and randomly weights pruning with different retraining schemes.

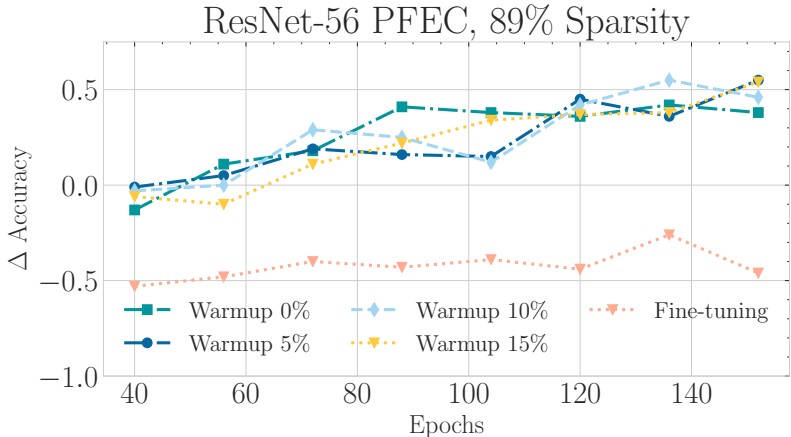

Figure 8: $\ell_1$-norm Filters Pruning (Li et al., 2016) on CIFAR-10 with different number of warmup epochs.

## F    PERFORMANCE OF CLR WITH DIFFERENT WARMUP EPOCHS

To avoid the laborious hyperparameters tuning process, we simply warm up the learning rate for CLR and SLR for the first $10\%$ budget of retraining. In this section, we investigate whether the performance of pruned network is sensitive to this hyperparameter.

Figure 8 presents the results of pruned networks obtained with fine-tuning and CLR while varying number of warmup epochs namely $0, 5, 10, 15\%$ of total retraining. We can observe that there is no significant diversity in results of CLR with different number of warmup epochs and all of them exceed performance of fine-tuning by a large margin.

## G    TRAINING FROM SCRATCH WITH CLR

In this section, we conduct experiment where we train the unpruned models from scratch with CLR schedule. Then, we compare performance of networks pruned from these baseline with those pruned from models trained with conventional step wise learning rate schedule. Table 14 presents the accuracy of pruned ResNet-56 models on CIFAR-10 with different training and retraining recipes. We use the budget of 160 for training the baseline networks. For retraining the trimmed networks, we examine the budget of 40 and 56 epochs. We can see that both CLR and SLR consistently exceed the performance of fine-tuning under these configurations.

Table 14: Performance of ResNet-56 pruned from models trained with CLR and conventional step-wise learning rate schedule on CIFAR-10. "Epochs" column indicates the number of epochs for retraining trimmed networks. "Schedule" colum indicates the learning rate schedule used for training baseline (unpruned) networks. The best and second best methods are highlighted in **bold blue** and blue respectively.

| Epochs | Schedule | Param $\downarrow$ % | Baseline | Fine-tuning | SLR | CLR |
|--------|----------|------------|----------|-------------|-----|-----|
| 40 | Step-wise | 13.7 | $93.15 \pm 0.36$ | $92.81 \pm 0.49$ | $92.93 \pm 0.15$ | $\mathbf{93.14 \pm 0.37}$ |
| | CLR | 13.7 | $93.45 \pm 0.17$ | $93.06 \pm 0.20$ | $93.03 \pm 0.16$ | $\mathbf{93.15 \pm 0.11}$ |
| | Step-wise | 34.9 | $93.15 \pm 0.36$ | $92.25 \pm 0.35$ | $\mathbf{92.83 \pm 0.16}$ | $92.81 \pm 0.32$ |
| | CLR | 34.9 | $93.45 \pm 0.17$ | $92.50 \pm 0.07$ | $92.63 \pm 0.05$ | $\mathbf{93.03 \pm 0.16}$ |
| 56 | Step-wise | 13.7 | $93.15 \pm 0.36$ | $92.81 \pm 0.41$ | $92.86 \pm 0.28$ | $\mathbf{93.22 \pm 0.37}$ |
| | CLR | 13.7 | $93.45 \pm 0.17$ | $93.04 \pm 0.16$ | $93.22 \pm 0.05$ | $\mathbf{93.37 \pm 0.25}$ |
| | Step-wise | 34.9 | $93.15 \pm 0.36$ | $92.48 \pm 0.45$ | $92.89 \pm 0.11$ | $\mathbf{93.26 \pm 0.17}$ |
| | CLR | 34.9 | $93.45 \pm 0.17$ | $92.63 \pm 0.13$ | $92.83 \pm 0.03$ | $\mathbf{93.29 \pm 0.25}$ |

