# OpenReview forum: "Network Pruning That Matters:  A Case Study on Retraining Variants"
_ICLR.cc/2021/Conference — ICLR 2021 Poster_

### Official Review · AnonReviewer1 · 2020-10-27
**initial review**

**Rating:** 6
**Confidence:** 3

**Review:**

The paper conducts extensive experiments to understand the reason behind the uncanny effectiveness of learning rate rewinding: the usage of a large learning rate.

### pros
* The paper, in general, is well-written, and the main message is very clear.
* The paper identifies the reason behind the success of learning rate rewinding through several aspects: retraining cost, model size, and pruning algorithms. It provides guidelines alternative to fine-tuning for practitioners to obtain compact models with better performance after network pruning.
* The observations of the random pruning are interesting and are aligned with the prior work [1].

### cons
1. The paper needs to improve its clarity.
    * It is suggested to include the details for the scores (e.g. 1.12 x in the title) of the subfigures in Fig. 2. Right now it is unclear to me how to calculate the value and why it is meaningful to present the results under these values.
    * Can you justify why it is necessary to include the learning rate warmup scheme for SLR and CLR, and why the paper only uses 10% of the total retraining budget? How much will the different fractions of the warmup epochs impact the retraining performance? An ablation study is required here.
    * The observations for random pruning with learning rate restarting are interesting but more details are required. E.g., it is unclear to me the form of performing the random pruning. Is it layerwise random pruning (same sparsity per layer) or global-wise random pruning?
2. Potential unfair comparison by using CLR.
    * The paper investigates the impact of different learning rate schedules for the re-training, after pruning on the model pre-trained by the standard stage-wise learning rate schedule. This design choice is sufficient to provide some practical guidelines, but it may also blur the contribution: as CLR is quite different from the other learning rate schemes, it is natural to question if the performance gain is solely due to a better learning rate schedule (but not large learning rate). Can you also provide an ablation study in terms of using CLR for both the training from scratch and re-training, and then compare both the accuracy and the accuracy drop scores?
    * The paper demonstrates the efficacy of CLR in terms of re-training the pruned model on the standard image classification benchmark. However, it is unclear to me if the same observations can be generalized to other CV tasks or even NLP tasks. It is encouraged to include some preliminary results to argue the generalization ability of the provided practical guidelines.

### reference
1. Sanity-Checking Pruning Methods: Random Tickets can Win the Jackpot, NeurIPS 2020.

### post-rebuttal
The authors have addressed most of my concerns, thus I will increase my score from 5 to 6.

---

> ### Author Response · Authors · 2020-11-19
> **Thank for your kind response. Here are our answers.**
>
> **“The details for the scores (e.g. 1.12 x in the title) of the subfigures in Fig. 2”**:
> In fact, this is the compression ratio of the pruned network (i.e. #param_before_pruning / #param_after_pruning). The reason we report the performance of these is that we adopt the pruning configuration of PFEC.
>
> **“Why it is necessary to include the learning rate warmup scheme for SLR and CLR, and why the paper only uses 10% of the total retraining budget? How much will the different fractions of the warmup epochs impact the retraining performance? An ablation study is required here.”**:
> As we just retrain a pretrained network, we adopt the common heuristic of using warming up learning rate. We would say that 10% budget for retraining is an arbitrary choice. Thus, we also do an ablation study with different number of warming up epochs in Appendix F.1. In summary, we do not find any noticeable difference between the performance of these settings and all of them consistently achieve higher accuracy than fine-tuning.
>
> **Random pruning type**: We have clarified this detail in the paper. See Paragraph 2 in Section 4.2.
> “Potential unfair comparison by using CLR” and “CLR and the performance gain”: In the paper, we had an experiment that we compare CLR to SLR. In fact, both CLR and SLR consistently outperform fine-tuning in our experiments. So learning rate schedule matters.

---

> > ### Comment · AnonReviewer1 · 2020-11-20
> > **post-rebuttal**
> >
> > Thank you for adding more (and extensive) results in the submission.
> >
> > Your responses have addressed most of my concerns, except the comparison between CLR and SLR:
> > 1. Which table/figure are you referring to, in terms of "In the paper, we had an experiment that we compare CLR to SLR."?
> > 2. Do you have any results comparing (1) CLR (training from scratch) and (2) CLR (re-training after the pruning on the model from (1))? A pointer would be very helpful.

---

> > > ### Author Response · Authors · 2020-11-23
> > > **Added more experiments for ablation study**
> > >
> > > Thanks for your response. We have amended the paper to address your concern as below:
> > >
> > > **Which table/figure are you referring to, in terms of "In the paper, we had an experiment that we compare CLR to SLR."?**: Through our extensive experiments with both CLR, SLR, Fine-tuning (as presented in Figure 2,3,4) we found that the large learning rate exceeds the performance of fine-tuning especially with high compression ratio and higher retraining budgets.
> > >
> > > **Do you have any results comparing (1) CLR (training from scratch) and (2) CLR (re-training after the pruning on the model from (1))? A pointer would be very helpful.** Thanks for your interesting suggestion, we have added Section G of the Appendix of the revised paper to address your concern. Specifically, we observed that CLR and SLR also outperform fine-tuning regardless of the learning rate schedule of baseline models.
> > >
> > > We also added some experiments in Section C of the Appendix to demonstrate the effectiveness of CLR for Taylor Pruning with other compression ratios.
> > >
> > > Thank you.

---

### Official Review · AnonReviewer2 · 2020-10-28
**Paper that explores learning rate schedules when re-training after pruning, and shows that re-training learning rate can matter more than pruning saliency metric.**

**Rating:** 6
**Confidence:** 5

**Review:**

# Summary

This paper analyzes the role of learning rate in re-training after pruning, building on previous findings that changing the learning rate schedule of re-training can result in higher accuracy than low-learning-rate fine-tuning. The paper proposes several learning rate schedules to compare, specifically a cyclic learning rate (gradually ramping up to and back down from the maximum learning rate schedule used during the original training phase) and a compressed version of the original learning rate schedule, and shows that these learning rate schedules outperform standard fine-tuning and also learning rate rewinding, showing that the findings of prior work come from using a higher learning rate in general and not any specific schedule. The paper than shows that choice of re-training learning rate schedule can have more impact on final accuracy than choice of saliency metric.

# Strengths

The paper, fairly conclusively, finds the following novel results:
- Re-training with a cyclic learning rate outperforms learning rate rewinding, seemingly leading to a new state-of-the-art re-training algorithm
- Re-training with a scaled learning rate schedule attains similar accuracy to re-training with learning rate rewinding
- Re-training a network that has already been pruned and trained with some scheme (e.g., Soft Filter Pruning, Taylor expansions) can reach higher accuracy with cyclic learning rate re-training than standard fine-tuning re-training.

Less conclusively, though still with reasonable evidence, the paper finds:
- The choice of learning rate schedule when re-training can have a higher impact on accuracy than the choice of saliency metric: specifically, at higher sparsities, re-training a randomly pruned netowrk with a cyclic learning rate schedule results in higher accuracy than re-training a magnitude pruned network with fine-tuning at a low learning rate.

# Weaknesses

- The evaluation of the paper focuses heavily on structured pruning. However, structured pruning can be an unreliable testbed for many of these techniques, as shown by [1]. The paper would benefit from discussion of [1] in relation to the structured pruning results -- for example, can similar accuracy be attained by training a randomly initialized pruned network with the same effective learning rate schedule? Without discussion or evaluation of baselines from [1], it's hard to know quite how to interpret the structured pruning results.
- The findings about the interplay between pruning saliency metrics and re-training schedule (Section 4), while interesting, are only minimally validated. Specifically, it would be interesting to see full curves of accuracies for each of these techniques like the curves in Figure 4 (at least for weight-level pruning) with a plot showing the accuracy of ResNet-56 with "Random Pruning + Fine-tuning", "Magnitude Pruning + Fine-tuning", "Random Pruning + CLR", "Magnitude Pruning + CLR" across different sparsities, which would generate a lot more confidence in this result
- The paper lacks specific hypotheses which are tested and validated/falsified. Specifically -- it's hard to know what conclusion to pull from Sections 3.1 and 3.2 other than that CLR > SLR >= LRW, and it's hard to know what conclusions to draw from Section 3.3.
- The more conclusive findings of the paper, that a high learning rate is important for optimization of pruned networks and that cyclic learning rates improve on learning rate rewinding, are a relatively incremental contribution


# Overall recommendation

5: Weak reject

I would be willing to raise this score if the authors address some of the weaknesses listed above. Specifically, if the authors can demonstrate that the results on MWP in Table 3 consistently generalize to other sparsities, or generate more confidence in the structured pruning findings (e.g., by showing that they result in higher accuracy than the trained-from-scratch structured pruned networks in [1]).

# Other comments and suggestions

- Minor typo in Figure 2 caption: "LLRW"
- HRank in Section 3.3: if the results are not presented and discussed in the main body of the paper, it'd probably be better to move the entire discussion of HRank to the appendix
- what does "Params" in the tables mean? I assume it means sparsity (i.e., percentage of parameters that are pruned away), but this is never made explicitly clear.
- why is R-FT missing for MWP in Table 3?

# References used in review:

[1] Zhuang Liu, Mingjie Sun, Tinghui Zhou, Gao Huang, and Trevor Darrell. "Rethinking the value of network pruning"


# Update post author response:

Thanks to the authors for the response. The newly reported results (specifically, those of Appendix E.2) satisfactorily address my concerns about both the generalization of the pruning method v.s. re-training scheme results, both in terms of sparsity levels and unstructured/structured pruning (though the observation of the relationship between R-CLR and FT do not hold quite as strongly for unstructured pruning, they do hold at high enough sparsities to be interesting). I’ve raised my score to a 6 as a result.

---

> ### Author Response · Authors · 2020-11-19
> **Thank for your detailed and constructing reviews. Here are our answers.**
>
> **"The findings of the interplay between pruning saliency metrics and re-training schedule (Section 4), while interesting, are only minimally validated"**: We have updated a few more experiments with other pruning algorithms (Taylor Pruning, the extension of HRank namely HRankPlus) in the revised version (Section 4.2).
>
> **"The paper lacks specific hypotheses which are tested and validated/falsified"**: we focused on empirical studies in this work that raise awareness of the implementation details for making fair comparisons in network pruning. Studying a hypothesis is left as future work.
>
> **"The more conclusive findings of the paper, that a high learning rate is important for optimization of pruned networks and that cyclic learning rates improve on learning rate rewinding, are a relatively incremental contribution"**: we agree with this perspective. The use of a large learning rate and different learning rate rewinding are often overlooked. And there could exist better learning rate schedules than what we used in this paper.
>
> **"The results on MWP in Table 3" and "structured pruning findings"**: We updated the results as requested. Please see Appendix E.2 and Section 3.4, respectively.

---

### Official Review · AnonReviewer4 · 2020-10-29
**Re-training matters as much as sparsification**

**Rating:** 8
**Confidence:** 5

**Review:**

The authors conducted a comprehensive set of experiments on choices of learning rate schedules for re-training/fine-tuning during iterative or after 1-shot pruning of deep convnets.  Empirically, they reported that high learning rate (LR) is particularly helpful in recovering generalization performance of the resultant sparse model.  The results are purely empirical, well-documented observations from well-designed experiments, which is of practical value in practice of network compression, and the consistent, somewhat surprising observation raises interesting questions.

Notably, this work has brought to attention an important but often overlooked aspect of network pruning: there exist complex interactions between the dynamics of optimization and sparsification, and as a consequence, it is only fair to compare two sparsification techniques when each of them are put in the _best_ optimization setup, respectively.

I have a few comments that I wish the authors would address here, discuss in revision or note for future work:

(1) Why is large LR helpful in recovering the accuracy of sparse nets?  There is little information provided in these experimental results to shed light on this question.  There has been loss landscape studies of sparse nets during training (such as arxiv:1906.10732, arxiv:1912.05671)--perhaps these could be applied to study the problem.  If the high LR's role were to knock the solution out of bad local minima, then does adding noise to gradients or smaller batch size achieve similar effect at the initial phase of re-training?

(2) Given a fixed re-training flop budget, after a pruning operation on the network, both (a) weight value rewinding (as in the Lottery Ticket Hypothesis training), (b) re-training LR schedule (as in this work) might be potentially helpful.  How does weight value rewinding interact with LR?

(3) For the random pruning results in Sec. 4, do fine-grain unstructured pruning methods present the same results?

(4) Does the result generalize to transformer models?  What about optimizers?  Does Adam present a same story as SGDM?

Page 5, line1 of the 3rd paragraph of Sec. 3.2: typo "reachs"

---

> ### Author Response · Authors · 2020-11-19
> **Thank for your constructing reviews and recommendation. Here are our answers.**
>
> **"Why is large LR helpful in recovering the accuracy of sparse nets?"**: We hypothesis that the flatness of the loss landscape might change substantially after pruning similar to quantization in [1] which might trap the pruned network in suboptimal solution. Moreover, the pruned network might fail to converge with just small number of retraining epochs, thus, large learning rate helps increase the convergence speed of trimmed networks in these cases. However, a rigorous theoretical explanation is left for future work.
>
> **"How does weight value rewinding interact with LR?"**: We further elaborated the interplay of pruning algorithms and learning rate schedule in Section 4.1. We leave the interesting idea of using weight value rewinding as future work.
>
> **"For Sec. 4, do fine-grain unstructured pruning methods present the same results?"**: we present the results of (iterative and oneshot) random pruning with MWP in Appendix E.2
>
> Reference:
>
> [1] HLHLp: Quantized Neural Networks Training for Reaching Flat Minima in Loss Surface, AAAI 2020.

---

### Official Review · AnonReviewer3 · 2020-10-29

**Rating:** 5
**Confidence:** 4

**Review:**

## Summary
This work focuses on evaluating different fine-tuning strategies after structured and unstructured pruning. The results show that high learning rate schedules (like cosine schedule) attain best performance in many different setting. With this high learning rate random (structured) pruning seems to work as well as the other pruning criteria. Overall the work has a strong coverage of experiments and the results could be helpful to the community. However, I think the work misses some important baselines and require a bit more work on writing.

## Pros
- A comprehensive coverage of different pruning algorithms is definitely a plus. Experiments are focusing more on the structured pruning methods, which I am not sure useful given the results of [1] (see cons below).

- Authors seem to have a good knowledge of recent structured pruning methods, which makes the study and the experiments convincing/strong.

## Cons
- I don't think the following statement is true (at least many of the unstructured pruning methods: "In most cases, pruning consists of three steps: training..prune..retrain" The paper starts with this premise and ignores many of the other iterative pruning methods (which prune during training). Many of the best unstructured pruning methods are iterative GMP (https://arxiv.org/abs/1710.01878), DNW (https://arxiv.org/pdf/1906.00586.pdf), STR (https://arxiv.org/abs/2002.03231).

- Only exception to the point above is the Table-1 (SFP); however in that table the authors miss an important baseline which is scaling the original training proportional to the Training+FT like it is done in [1, 2, 3]. This baseline should be added to the table and considered wherever possible.

- "Recently, Renda et al. (2020) proposed a state-of-the-art technique for retraining pruned networks namely learning rate rewinding" I don't think this sentence is accurate. It's true authors claim SOTA, but I would argue they miss an important baseline in which the original training schedule of the iterative pruning algorithms are scaled according to the training budget as it is done in [1, 2, 3]. In my experience this method performs better than LRW. Therefore it would be nice to include this baseline whenever possible.

- Does CLR results for l1-pruning exceed Scratch-B results of [1]. If yes, this is very important to mention/highlighy. Otherwise, I like to see a discussion about why we should care about structured pruning methods as the main focus of the work seems to be those.

- What is the difference between PFEC and PFEC-B? And why do authors use this acronym? "l1-structured" might be a more appropriate choice. And what are the multipliers in Figure-2 (i.e. 1.12x, 1.45x...)? It would be better to use "sparsity".

- "A well-known practice is fine-tuning, which aims to train the pruned model with a small fixed learning rate." and "More advanced learning rate schedules exist, which we generally refer to as retraining." I rather call of them fine-tuning or warm-restart, as all networks start with a pretrained network. This terminology would align better with the previous work. Then you call call constant-lr, lrw, clr, etc...

## Minor Points
- "Here we hypothesize that the initial pruned network is a suboptimal solution, staying in a local minima." I found this statement a bit vague. Networks are usually not not converged after they are pruned (or even at the end of an ImageNet training). We also don't know whether with long enough training the small learning rate finetuning would get same good results or not. It would be nice to make this statement more precise. Maybe something like "high learning rate would help find better minima faster."
- "as `1-norm filters pruning" filter pruning. Structured pruning is used more often similarly "weights pruning" -> unstructured pruning
"For simplicity, we always use the largest learning rate of the original training for learning rate restarting..." I didn't understand this statement. Is this for CLR?
- "For CLR and SLR, the learning rate is increased from the smallest learning rate of original training to the largest one according to cosine function" probably the other way around Learning rate is decayed over time?

- "For ImageNet, we run each experiment once." It would be great to run few more to get more precise results before the final version.

- "under both setting learning rate restarting approaches consistently..." -> under both settings lr-restart consistently...

- Is Figure-3 a/b MWP? It would be nice to mention this in the title or caption.

- "Being that said," -> That being said

- "there are notable differents between" ->  differences

- "for future works." -> for future work

[1] RETHINKING THE VALUE OF NETWORK PRUNING, https://arxiv.org/pdf/1810.05270.pdf
[2] The State of Sparsity in Deep Neural Networks, https://arxiv.org/abs/1902.09574
[3] Rigging the Lottery: Making All Tickets Winners, https://arxiv.org/pdf/1911.11134.pdf

## After Rebuttal
- I thank authors for considering my suggestions. I increase my score to 5. Having a quick look (I am sorry that I didn't have more time) at the new results; most results on structured pruning seem to agree with [1]; with some improvements over baselines when CLR is used when training from scratch. Results on unstructured pruning seems minimal and focuses mostly on one-shot pruning; and furthermore the baseline suggested above (i.e. scaling the entire learning_rate schedule)  is not added to the iterative pruning results. Overall, I like the direction of the paper, but I think the motivation should be improved and results should be distilled.

---

> ### Author Response · Authors · 2020-11-19
> **Thank you for your insigntful comments. Here are our answers.**
>
> **"The difference between PFEC and PFEC-B? And why do authors use this acronym? And what are the multipliers in Figure-2 (i.e. 1.12x, 1.45x...)?"**: 1.12x denotes the compression ratio (#parameter_before_pruning / #parameter_after_pruning) of pruned networks. We use these compression ratios as in the original paper of Li et al., 2016 [1]. To improve the clarity of the paper we have changed this value to sparsity in the revised version according to the request of the reviewer.
>
> **"For simplicity, we always use the largest learning rate of the original training for learning rate restarting... I didn't understand this statement. Is this for CLR?"**: Since we restart the learning rate to a (relatively) high value in both CLR and SLR, we employ learning rate warming up for both retraining schemes. Specifically, the learning rate is increased from the smallest value during training to the highest value in the first 10% retraining epochs. We also conducted an ablation study that varies the number of warming up in Appendix F.1. We found that the final performance is not sensitive to this choice.
>
> **"For ImageNet, we run each experiment once."**: We have tried to run ImageNet three times in Section 3.4. Beyond that, we ran experiments on ImageNet with a wide range of pruning algorithms as shown in Section 4.
>
> Reference
>
> [1] Pruning Filters for Efficient ConvNets, ICLR 2017.

---

### Author Response · Authors · 2020-11-19
**Revision update and response to common questions**

We are grateful to the reviewers for providing constructive feedback. All reviewers agreed that our experiments are comprehensive, and "the results could be helpful to the community" (Reviewer 3), having "practical value in practice of network compression, and the consistent, somewhat surprising observation raises interesting questions." (Reviewer 4), being "fairly conclusively" with novel results (Reviewer 2), and "the observations of the random pruning are interesting and are aligned with the prior work" (Reviewer 1). The reviewers also raised several issues that we have attempted to consider and address. We have already revised the paper accordingly. Please find our responses below.



### Revision Summary
- We added a new Section 3.4 to include comparison between CLR and the requested baseline (i.e. scaling scratch training) as suggested by Reviewer 2 and 3. Some results in Section 3.3 is shorten and shifted to Appendix.
- We added Section 4.1 to validate the finding of "the importance of retraining" by providing evidence that PFEC+CLR (*without hyperparameter tuning*) can very well surpass other complex pruning algorithms with the *same* retraining budgets and compression ratio (while the architectures are **not** necessarily the same). Specifically, we compare PFEC+CLR with Taylor Pruning [1], GAL [2], HRankPlus (extension of HRank[3]) in Section 4.1 and Discrimination-aware Channel Pruning [4], Provable Filters Pruning [5] in Appendix D.2.
- In Section 4.2., we updated the comparison of random pruning with more methods, i.e., we added Taylor Pruning on ImageNet and HRankPlus (an extension of HRank which is more efficient and effective) on CIFAR-10. See Table 5.
- We improved the clarity of the writing based on the suggestion from the reviewers. The revised text is highlighted in blue.

### Common questions
- **Extra baselines**: Reviewer 2 and 3 suggested that comparison to the baseline by Liu et al. [6] is necessary. We added two new baselines, Scratch-B and Scratch-E that randomly initialize and train a pruned network with a fair budget compared to other methods, to a new Section 3.4 accordingly. In our experiments, it can be seen that Scratch-B and Scratch-E outperforms fine-tuning, which aligns to the result by Liu et al. [6]. More importantly, CLR has similar performance to such baselines on CIFAR, and outperforms these baselines on ImageNet. Our result demonstrates an example that structured pruning can be effective and worth more future investigations.
- **Generalization to NLP tasks**: In principle, our findings should generalize to NLP tasks, but running the empirical studies and analysis in this domain deserves a separate future work.

**Reference**

 [1] Importance Estimation for Neural Network Pruning, CVPR 2019.

 [2] Towards Efficient Model Compression via Learned Global Ranking, CVPR 2020.

 [3] HRank: Filter Pruning using High-Rank Feature Map, CVPR 2020.

 [4] Discrimination-aware Channel Pruning for Deep Neural Networks, NeurIPS 2018.

 [5] Provable Filter Pruning for Efficient Neural Networks, ICLR 2020.

 [6] Rethinking the Value of Network Pruning, ICLR 2019.

---

### Decision · Program_Chairs · 2021-01-07
**Final Decision**

**Decision:**

Accept (Poster)

**Comment:**

This paper follows the observations of Renda et al. (2020) that the learning rate in the fine-tuning or retraining phase of neural network pruning is an under-considered component of the pruning process. Renda et al. (2020) argue for a technique that uses the learning rate schedule of the original training regime for fine-tuning. However, their work does not offer a hypothesis or an explanation for why this works.

This work instead offers more insight into why reusing the original learning rate is productive. Specifically, it shows that using high learning rates is the key component. To demonstrate this, the paper includes a study of using the original step-wise learning from the original training regimen, except accelerated for a given number of fine-tuning epochs. The paper also demonstrates that Cyclic Learning Rate Restarting (CLR) also provides an effective, if not better, learning rate schedule for fine-tuning.

As noted by the reviewers, the core observations and contributions of this work are modest, but are still a valuable addition to the literature in the pruning community.

Having said that, there are some confounding issues with CLR. Specifically, that CLR itself may simply be a more effective learning rate schedule for training neural networks, independent of the particular application to fine-tuning (Reviewer 1). The revision includes an additional appendix that dispels some of this concern. However, indeed, the CLR does improve the base network performance for some configurations.

Broadly, the value proposition here is a thorough demonstration of learning rate schedules for fine-tuning with an overall take that comparisons between techniques need be more sensitive to this choice as previous work perhaps has not thoroughly considered alternative learning rates.